# Impacts of Urban Green on Cardiovascular and Cerebrovascular Diseases—A Systematic Review and Meta-Analysis

**DOI:** 10.3390/ijerph20115966

**Published:** 2023-05-26

**Authors:** Alessandro Bianconi, Giulia Longo, Angela Andrea Coa, Matteo Fiore, Davide Gori

**Affiliations:** Department of Biomedical and Neuromotor Sciences, University of Bologna, 40126 Bologna, Italy

**Keywords:** urban green, cardiovascular diseases, ischemic heart disease, cerebrovascular diseases, residential greenness, CVD, meta-analysis

## Abstract

Cardiovascular diseases (CVDs) are a leading cause of mortality globally. In particular, ischemic heart diseases (IHDs) and cerebrovascular diseases (CBVDs) represent the main drivers of CVD-related deaths. Many literature examples have assessed the association between CVD risk factors and urban greenness. Urban green (UG) may positively affect physical activity, reduce air and noise pollution, and mitigate the heat island effect, which are known risk factors for CVD morbidity. This systematic review aims to assess the effects of urban green spaces on CVD morbidity and mortality. Peer-reviewed research articles with a quantitative association between urban green exposure variables and cardiovascular and cerebrovascular outcomes were included. Meta-analyses were conducted for each outcome evaluated in at least three comparable studies. Most of the included studies’ results highlighted an inverse correlation between exposure to UG and CVD outcomes. Gender differences were found in four studies, with a protective effect of UG only statistically significant in men. Three meta-analyses were performed, showing an overall protective effect of UG on CVD mortality (HR (95% CI) = 0.94 (0.91, 0.97)), IHD mortality (HR (95% CI) = 0.96 (0.93, 0.99)), and CBVD mortality (HR (95% CI) = 0.96 (0.94, 0.97)). The results of this systematic review suggest that exposure to UG may represent a protective factor for CVDs.

## 1. Introduction

The proportion of the population living in urban settings has grown rapidly in recent decades. By 2050, 68% of the world’s population is expected to live in urban areas [1]. With the increase in population living in an urban setting, more and more people will be exposed to cities’ environmental factors, which could impact human health [2].

In recent history, the urbanization process has been addressed as an ameliorative process for the living conditions of residents, providing more excellent job opportunities, improved socio-economic status (SES), and greater access to healthcare. However, in recent years it has become increasingly evident and pressing how certain aspects of urban living (such as air pollution, noise, and extreme heat) can negatively affect the quality of life of the citizens [3].

The design of built environment elements has been shown to influence physical and mental health [4]. Urban living may hinder access to the natural environment and increase exposure to harmful environmental hazards such as air pollution and heat [5]. Some studies have suggested that urban green (UG), defined as all urban land covered by vegetation of any kind, may impact air quality and traffic noise and help mitigate the temperature rise due to climate change in urban settings [6,7]. Furthermore, the presence of a natural environment might encourage physical activity among people living nearby, resulting in several health benefits [8].

Cardiovascular diseases (CVDs) are the leading cause of death globally, accounting for more than 18 million deaths per year, with ischemic heart diseases (IHDs) and cerebrovascular diseases (CBVDs) representing the main drivers of CVD-related deaths [9]. Many modifiable risk factors linked to CVDs have been shown to be connected to green areas in the urban environment. The implementation of urban green spaces can positively affect physical activity, air pollution, noise, and heat [7], which are well-known risk factors for cardiovascular and cerebrovascular morbidity [9]. Therefore, the lack of UG in an urban environment may pose a threat to citizens’ cardiovascular and cerebrovascular health. Literature examples have found associations between residential greenness and CVD risk factors, such as hypertension and diabetes [10]. Moreover, studies have shown that a higher amount of greenness in a residential area is associated with a lower mortality rate [11]. There is growing evidence in the literature about the associations between green space exposure and cardiovascular and cerebrovascular health, but no clear effect of UG exposure on CVDs has been stated yet.

The aim of this systematic review of the literature is to assess the impacts of UG on cardiovascular and cerebrovascular disease morbidity and mortality.

## 2. Materials and Methods

### 2.1. Study Design

The PRISMA statement [12] guidelines were followed while conducting this systematic review. A comprehensive and systematic literature search was performed through PubMed and Scopus databases. The search strategy was adapted for each database. Keywords referring to urban green exposure, cardiovascular, heart, and cerebrovascular diseases were used in composing the search string. The final search strings for each database were the following, respectively, for Pubmed and Scopus:

(((Urban[tiab] OR residential[tiab]) AND (green*[tiab] OR forest[tiab] OR park*[tiab])) OR “green areas” OR “green zones” OR “green spaces”) AND (stroke[tiab] OR cerebrovascular[tiab] OR “Cerebrovascular Disorders”[Mesh] OR cardiovascular[tiab] OR “Cardiovascular Diseases”[Mesh]).

((TITLE-ABS-KEY (“urban” OR “residential”) AND TITLE-ABS-KEY (“green” OR “greenness” OR “forest” OR “park”)) OR TITLE-ABS-KEY (“green areas”) OR TITLE-ABS-KEY (“green zones”) OR TITLE-ABS-KEY (“green spaces”)) AND TITLE-ABS-KEY (“stroke” OR “cerebrovascular” OR “cardiovascular” OR “infarction” OR “ischemic”).

The last search was conducted on 18 July 2022. The review protocol was registered on PROSPERO on 13 July 2022 (ID: CRD42022346328).

### 2.2. Study Eligibility Criteria and Selection Strategy

Peer-reviewed studies were considered, with no restrictions on the geographic origins of the articles. However, only studies published in the English language were included.

The following inclusion criteria were applied during the selection process:Primary/original research articles;Studies involving humans;Studies with a quantitative association between urban green exposure variables and cardiovascular and cerebrovascular outcomes.

The following exclusion criteria were also applied during the selection process:
Meta-analyses and reviews;Non-research articles;Non-primary research articles;Studies involving animals;Articles in languages other than English;Studies with no correlation assessed between urban green and cerebrovascular or cardiovascular outcomes.

Two researchers selected the studies following the inclusion criteria independently and blindly. AB, AAC, MF, and GL participated equally in the selection process. Each study was randomly assigned to two researchers. Discordances in the inclusion process were resolved by discussion between the two reviewers. In case of persistence of the discordance, a third reviewer acted as an external and unique arbiter.

The first article selection process analyzed the titles and the abstracts only, considering pertinence. The second process consisted of a full-text analysis of the eligible studies to further assess their inclusion appropriateness and pertinent data availability.

A quality assessment of the selected articles after the full-text screening was performed, assigning a quality score for each article (Table 1). Each article was assessed by two authors independently. AB, AAC, MF, and GL participated equally in the quality assessment process. Each study was randomly assigned to two researchers. Disagreement in the assessment process was resolved by consensus between the two authors involved. As a consequence of the included studies’ heterogeneity, the current literature was searched in order to find an existing quality score suitable for this review. The score used by Gascon et al. was selected as the most suitable [11]. This score, ranging from 0 to 100, was used to evaluate the quality of the included studies on the basis of the following eight items: study design, confounding factors, statistics, potential bias, multiplicity, green exposure assessment, effect size, and participants have been living at least 1 year in the studied area [11].

After discussion among all the authors, a template for data extraction was chosen including title, journal, author(s), year of publication, country, study design, study population, sample size, urban green exposure variable (i.e., the quantity of green coverage at residential level), outcome variable (i.e., general CVD or specific disease prevalence or mortality), type of estimate and estimate of association, and confounding variables (Table 1 and Appendix A).

### 2.3. Meta-Analysis

Meta-analyses were performed for each outcome evaluated in at least three studies with matched study design, choice of exposure variable, and effect measure. Statistical heterogeneity was tested in order to determine if it was appropriate to combine the studies for meta-analysis. Heterogeneity was calculated using the I2 statistic, and resulting values greater than 40% were considered substantially heterogeneous, according to the Cochrane Handbook for Systematic Reviews of Interventions [13]. In cases where the number of studies was less than 5 and/or studies were substantially heterogeneous, a random-effects model was always chosen [13]. Forest plots were created to display effect estimates with 95% CIs for individual studies’ effect estimates and pooled results. Sensitivity analyses were conducted for each meta-analysis. Funnel plots were created in order to assess the presence of publication bias. Data analyses were performed using RevMan Software (Review Manager, Version 5.4, The Cochrane Collaboration, Copenhagen, Denmark, 2020).

## 3. Results

Applying the search queries, a total of 349 articles were selected in PubMed and 470 articles in the Scopus database. A total of 221 articles were removed after duplicate comparison. After screening the titles and abstracts, 529 articles were excluded. A total of 69 articles were assessed for eligibility through full-text evaluation of which 36 were included in the study (Figure 1) [10,14,15,16,17,18,19,20,21,22,23,24,25,26,27,28,29,30,31,32,33,34,35,36,37,38,39,40,41,42,43,44,45,46,47,48].

The year of publication of the included studies ranged between 2012 and 2022, and only five studies were published before 2020. Most of the studies were conducted in East Asia (N = 11), North America (N = 9), Europe (N = 8), and Oceania (N = 5); only a few studies were conducted in South America (N = 2) and South Asia (N = 1). No studies were conducted in Africa, Central America, and other macro-regions of Asia.

Fifteen of the studies included were ecological studies, thirteen were cohort studies, seven were cross-sectional, and one was a case-control study. Quality score assessments ranged between 20% and 90%, with a median score of 60%.

The variables of exposure to UG chosen in the included studies were heterogenous, with the normalized difference vegetation index (NDVI) used in twenty-one studies, representing the most frequent variable. NDVI is calculated starting from satellite images produced by sensors that acquire data in the visible red and near-infrared. It evaluates photosynthetic activity, and it represents the main satellite indicator of the presence of vegetation on the Earth’s surface and its evolution over time. Other UG exposure variables, such as the percentage of green areas at the residential level or the leaf area index (LAI), are reported in Table 1.

The most evaluated outcome was CVD mortality in eighteen included studies; eight studies evaluated IHD mortality and six studies evaluated CBVD/stroke mortality. Moreover, eight studies evaluated CVD morbidity, prevalence, or incidence, and six studies evaluated stroke morbidity, prevalence, or incidence. Other outcomes evaluated in the included studies were IHD morbidity, prevalence, or incidence (N = 4); AMI morbidity, prevalence, or incidence (N = 3); HF morbidity, prevalence, or incidence (N = 1); CVD hospital admission (N = 2); first AMI hospital admission (N = 1); ischemic stroke morbidity (N = 2); hemorrhagic stroke morbidity (N = 2); CHD and stroke combined prevalence (N = 1); and CHD and stroke combined hospital admission (N = 1).

The results of each study included are summarized in Table 1.

**Table 1 ijerph-20-05966-t001:** Data extraction results. Abbreviations: CAU = census area unit; IQR = interquartile range; T3 vs. T1 = third tertile vs. first tertile; Q4 vs. Q1 = fourth quartile vs. first quartile; Q5 vs. Q1 = fifth quintile vs. first quintile.

Authors	Year	Country	Study Design	Population	Sample Size	UG Exposure Variable	Comparison	Outcome Variable	Type of Estimate		Estimate of Assosiation	Quality Score
Richardson et al. [14]	2010	UK	Ecological study	General population (aged 16–64)	Not specified, approximately 28.6 million	% Green space at CAU level	Four equal groups: >25%, 25–50%, 50–75%, >75% (highest vs. lowest)	CVD mortality	IRR (95% C.I.)			60
										Male	0.95 (0.91–0.98)	
										Female	1.00 (0.95–1.06)	
Richardson et al. [15]	2010	New Zeland	Ecological study	General population	1,546,405	% Total and % usable green space at CAU level	Q4 vs. Q1	CVD mortality	IRR (95% C.I.)			60
										Total green space	1.01 (0.91, 1.11)	
										Usable green space	0.96 (0.90, 1.03)	
Richardson et al. [16]	2012	USA	Ecological study	General population	Not specified, approximately 43 million	% Green space at CAU level	T3 vs. T1	CVD mortality	linear regression coefficient (β) (95% C.I.)			40
										Male	6.49 (−62.46 to 75.45)	
										Female	1.90 (−41.96 to 45.76)	
Pereira et al. [17]	2012	Australia	Cross-sectional	General population (>18 yrs old)	11,404	NDVI with 1600 m buffer from residence address	T3 vs. T1	CHD + Stroke prevalence, self-reported	OR (95% C.I.)		0.84 (0.68, 1.03)	60
								CHD + Stroke hospital admission			0.63 (0.43, 0.92)	
Villeneuve et al. [18]	2012	Canada	Cohort Study	General population (>35 yrs old)	Not specified, approximately 575,000	NDVI with 500 m buffer from residence address	IQR	CVD mortality	HR (95% C.I.)		0.94 (0.92–0.96)	90
								IHD mortality			0.96 (0.95–0.98)	
								Stroke mortality			0.96 (0.93–0.98)	
Richardson et al. [19]	2013	New Zeland	Cross-sectional	General population (>15 yrs old)	8157	% Green space at CAU level	Q3 vs. Q1 Q4 vs. Q1	CVD prevalence	OR (95% C.I.)		0.80 (0.64–0.99) [Q3 vs. Q1]; 0.84 (0.65–1.08) [Q4 vs. Q1]	70
Tamosiunas et al. [20]	2014	Lithuania	Cohort study	General population (aged 45–72)	5112	Residence distance from city parks larger than 1 hectare	T3 vs. T1	CVD	HR (95% C.I.)		1.36 (1.03–1.80)	70
										Male	1.51 (1.04–2.19)	
										Female	1.22 (0.79–1.89)	
Bixby et al. [21]	2015	UK	Ecological study	General population	Not specified, approximately 11 million people	% Green space of total city area	Q5 vs. Q1	CVD mortality	RR (95% C.I.)			50
										Male	0.95 (0.86–1.05)	
										Female	0.94 (0.83–1.07)	
Massa et al. [22]	2016	Brazil	Cross-sectional	General population (>60 yrs old)	1333	% Green space at CAU level	Q4 vs. Q1	CVD morbidity (self reported)	OR (95% C.I.)		0.48 (0.42–0.54)	50
Xu et al. [23]	2017	Hong Kong	Ecological study	General population	58,854 CVD-related deaths	NDVI at CAU level	IQR	CVD mortality	RR (95% C.I.)		0.88 (0.80, 0.98)	60
										Male	0.83 (0.74, 0.93)	
										Female	0.94 (0.84, 1.05)	
Wang et al. [24]	2017	China	Cohort Study	General population (>65 yrs old)	3544	% Green space in 300 m buffer from residence address	10% increase	CVD mortality	HR (95% C.I.)		0.888 (0.817 to 0.964)	90
								IHD mortality			0.912 (0.805 to 1.033)	
								Stroke mortality			0.658 (0.519 to 0.833)	
Crouse et al. [25]	2017	Canada	Cohort study	General population (≥19 yrs old)	1,265,515	NDVI with 250 m buffer from residence address	IQR	CVD mortality	HR (95% C.I.)		0.911 (0.894–0.928)	90
								IHD mortality			0.904 (0.882–0.927)	
								CBVD mortality			0.942 (0.902–0.983)	
da Silveira et al. [26]	2017	Brazil	Ecological study	General population	6,320,446	NDVI at CAU level	Q4 vs. Q1	IHD mortality	Coefficient of the Bayesian CAR model (β) (95% C.I.)		−0.069 (−0.101–−0.038)	50
								CBVD mortality			−0.048 (−0.083–−0.012)	
Vienneau et al. [27]	2017	Switzerland	Cohort study	General population (>30 yrs old)	4,284,680	NDVI in a 500 m buffer from residence address	IQR	CVD Mortality	HR (95% C.I.)		0.93 (0.92–0.94)	80
								IHD mortality			0.95 (0.93–0.97)	
								Stroke mortality			0.93 (0.89–0.96)	
Servadio et al. [28]	2019	USA	Ecological study	General population	Not specified, almost 6 million inhabitants	% Tree canopy cover at CAU level		CHD prevalence	Coefficient of the Bayesian CAR model (95% C.I.)		0.3600 (*p* < 0.05)	30
								Stroke prevalence			0.2012 (*p* < 0.05)	
Orioli et al. [29]	2019	Italy	Cohort Study	General population (>30 yrs old)	1,263,721	NDVI in 300 m from residence address	IQR	CVD mortality	HR (95% C.I.)		0.984 (0.975, 0.994)	90
								IHD mortality			0.985 (0.968, 1.001)	
								CBVD mortality			0.965 (0.944, 0.986)	
								Stroke incidence			0.976 (0.960, 0.993)	
Seo et al. [30]	2019	South Korea	Cohort study	General population	351,409	% Green space of total district area, limited to built environment	Q4 vs. Q1	CVD	HR (95% C.I.)		0.85 (0.81–0.89)	80
										Male	0.86 (0.80–0.92)	
										Female	0.85 (0.79–0.91)	
										Under 40 yrs old	0.88 (0.73–1.05)	
										40–60 yrs old	0.81 (0.75–0.87)	
										Over 60 yrs old	0.89 (0.84–0.95)	
								CHD			0.83 (0.78–0.89)	
								AMI			0.77 (0.68–0.88)	
								Total stroke			0.87 (0.82–0.93)	
								Ischemic stroke			0.86 (0.80–0.94)	
								Hemorrhagic stroke			0.98 (0.86–1.12)	
Wang et al. [31]	2019	USA	Ecological study	General population	1,530,981	% Green space at CAU level		CVD mortality	Negative binomial coefficient (95% C.I.)		−0.0041 (−0.0092, 0.0010)	60
Jennings et al. [32]	2019	USA	Ecological study	General population	335,327	% Tree canopy cover and LAI at CAU level		CVD hospital admission	OR (95% C.I.)			50
										Tree canopy cover	0.98 (0.97–1.01)	
										LAI	2.28 (0.91–5.74)	
Astell-Burt et al. [10]	2019	Australia	Cross-sectional	General population	46,786	% Total green space and % tree canopy in 1600 m buffer from residence address		CVD prevalence	OR (95% C.I.)			60
										Total green space	0.999 (0.996–1.002)	
										Tree canopy cover	0.996 (0.993–0.999)	
Kim et al. [33]	2019	South Korea	Ecological study	General population	317,869	NDVI at district level	IQR	CVD mortality	Percent changes in cause-specific mortality (95% C.I.)		−2.56% (−4.68%, −0.39%)	40
								IHD mortality			−3.45% (−6.84%, 0.07%)	
Paul et al. [34]	2020	Canada	Cohort study	General population (aged 35–85)	4,251,146	NDVI with 500 m buffer from residence address	IQR	Stroke morbidity	HR (95% C.I.)		0.96 (0.94–0.97)	90
Hartig et al. [35]	2020	Sweden	Ecological study	General population (>18 yrs old)	5,498,405	% Green space and % urban park at parish level		CVD mortality	IRR (95% C.I.)			50
										Green space	0.998 (0.995 to 1.000)	
										Urban park	1.001 (0.998 to 1.004)	
Astell-Burt et al. [36]	2020	Australia	Cohort Study	Type 2 Diabetes	4166	% of Tree canopy cover in 1600 m buffer from residence address	four intervals: <10% to 11–19.9%, 21–29.9%, or ≥30% (highest vs. lowest)	CVD mortality	HR (95% C.I.)		0.75 (0.47, 1.16)	80
								first CVD hospital admission			0.92 (0.77, 1.11)	
								first AMI hospital admission			0.77 (0.42, 1.36)	
Chen et al. [37]	2020	Canada	Cohort study	General population (>35 yrs old)	1,290,288	NDVI with a 250 buffer from residence address	IQR	CVD mortality	HR (95% C.I.)		0.91 (0.90–0.93)	90
								AMI incidence			0.94 (0.92–0.96)	
								HF incidence			0.95 (0.93–0.96)	
Yang et al. [38]	2020	China	Cross-sectional	General population (aged 18–74)	24,845	NDVI and SAVI with 500 m buffer from community centroid (not residential)		CVD prevalence	OR (95% C.I.)		0.73 (0.65–0.83)	70
											0.74 (0.66–0.84)	
Lee et al. [39]	2020	Taiwan	Ecological study	General population	Not specified	NDVI at township level		CVD mortality	RR (95% C.I.)		0.903 (0.791, 1.030)	50
Bauwelinck et al. [40]	2021	Belgium	Cohort Study	General population (>30 yrs old)	2,185,170	NDVI with 500 m buffer, % green space within buffer of 500 m	IQR	CVD mortality	HR (95% C.I.)			80
										NDVI	0.99 (0.97–1.01)	
										% Green space	1.01 (1.00–1.02)	
								IHD mortality				
										NDVI	1.02 (0.98–1.05)	
										% Green space	1.03 (1.01–1.05)	
								CBVD mortality				
										NDVI	0.99 (0.95–1.04)	
										% Green space	1.00 (0.98–1.03)	
Padmaka Silva et al. [41]	2021	Sri Lanka	Cross-sectional	Working-age men	5268	NDVI with 400 m from residence address		Heart disease, self reported	OR (95% C.I.)		0.80 (0.64, 1.00)	60
Liu et al. [42]	2021	China	Cross-sectional	General population	2100	NDVI with 1500 m buffer from residence address	T3 vs. T1	CVD prevalence	OR (95% C.I.)		0.618 (0.434–0.879)	70
										Male	0.768 (0.663, 0.890)	
										Female	0.906 (0.805, 1.020)	
										Under 65 yrs old	0.805 (0.669, 0.969)	
										Over 65 yrs old	0.836 (0.752, 0.930)	
Cheruvalath et al. [43]	2022	USA	Case-control study	General population (>18 yrs old)	5870 (1174 case and 4696 control patients)	NDVI with 250 m buffer from residence address	IQR	Stroke	OR (95% C.I.)		0.330 (0.111, 0.975)	70
								Ischemic stroke			0.32 (0.088–1.178)	
Wang et al. [44]	2022	China	Ecological study	General population	469 CVD deaths	NDVI at district level		CVD mortality	Spearman correlation coefficient ρ (*p*-value)		−0.179 (*p* < 0.01)	20
Li et al. [45]	2022	China	Ecological study	General population	6,334,875	NDVI at CAU level		IHD	Coefficient of the Bayesian CAR model (β) (95% C.I.)		−0.0044 (−0.0077, −0.0010)	40
Li et al. [46]	2022	China	Cohort Study	General population (adults)	32,521	NDVI in buffer of 250 m from residence address	IQR	IHD incidence	HR (95% C.I.)		0.89 (0.81, 0.98)	80
Ponjoan et al. [47]	2022	Spain	Cohort study	DM type 2 patients (>18 yrs old)	41,463	NDVI with buffer of 300 m around the census tract	0.01 increase	AMI	HR (95% C.I.)		0.94 (0.89–0.99)	90
										Male	0.91 (0.86–0.97)	
										Female	0.99 (0.92–1.08)	
Ho et al. [48]	2022	Hong-Kong	Ecological study	General population	8697 hemorrhagic stroke deaths and 10,270 non-hemorrhagic stroke deaths	NDVI at CAU level	Two groups: low exposure (mean NDVI < 0.1) vs. high exposure (mean NDVI ≥ 0.1)	Hemorrhagic stroke mortality	OR (95% C.I.)		0.961 (0.890, 1.037)	40
										Under 65 yrs old	0.983 (0.844, 1.145)	
										65 to 79 yrs old	1.006 (0.882, 1.148)	
										Over 89 yrs old	0.903 (0.801, 1.017)	
								Non-hemorrhagic stroke mortality			1.066 (0.993, 1.145)	
										Under 65 yrs old	1.101 (0.871, 1.391)	
										65 to 79 yrs old	1.091 (0.962, 1.237)	
										Over 89 yrs old	1.031 (0.940, 1.132)	

### 3.1. CVD Mortality

The eighteen studies analyzing the correlation between UG and cardiovascular mortality were heterogenous in study design and choice of UG exposure variable. Eleven studies found an inverse correlation between the amount of UG exposure and the risk of death by CVDs, and seven did not find any statistically significant correlation (Table 1).

Six cohort studies analyzing the correlation between NDVI around the residence address and CVD mortality were included in the meta-analysis. The median quality score of the included studies was 90%. The results show pooled HR of CVD mortality per IQR increase in NDVI, indicating an overall reduction in the risk of CVD mortality for each IQR increase in NDVI (HR (95% CI) = 0.94 (0.91, 0.97)) (Figure 2). NDVI buffers from the home address of each study were considered comparable by the authors, with all NDVI buffers ranging between 250 and 500 m from the residential address. The IQR of NDVI of the included studies ranged between 0.10 and 0.24. Sensitivity analyses were performed and showed no significant differences in the pooled results. The analysis of the funnel plot showed no evidence of publication bias. The results of the sensitivity analysis and funnel plots are reported in the Appendix A.

### 3.2. IHD Mortality

Eight studies analyzing associations between UG and IHD mortality were included. The study design and choice of UG exposure variable were heterogeneous. Four studies found a negative correlation between the amount of UG exposure and risk of death by IHD, three did not find any statistically significant correlation, and one study found no statistically significant correlation between NDVI and IHD mortality, but found a positive correlation between IHD mortality and the percentage of UG nearby the residence address (Table 1).

Five cohort studies analyzing the correlation between NDVI around the residence address and IHD mortality were included in the meta-analysis. The median quality score of the included studies was 90%. The results show the pooled HR of IHD mortality per IQR increase in NDVI, indicating an overall reduction in the risk of IHD mortality for each IQR increase in NDVI (HR (95% CI) = 0.96 (0.93, 0.99)) (Figure 3). NDVI buffers from the home address of each study were considered comparable by the authors, with all NDVI buffers ranging between 250 and 500 m from the residential address. The IQR of NDVI of the included studies ranged between 0.10 and 0.24. Sensitivity analyses showed that by excluding two studies (Vienneau et al. [27] and Villeneuve et al. [18]) from the meta-analysis, statistical significance was lost. The analysis of the funnel plot showed no evidence of publication bias. The results of the sensitivity analysis and funnel plots are reported in the Appendix A.

### 3.3. CBVD Mortality

Seven studies analyzing associations between UG and CBVD mortality were included. Six studies found a negative correlation between the amount of UG exposure and the risk of CBVD mortality, and one did not find any statistically significant correlation (Table 1).

Five cohort studies analyzing the correlation between NDVI around the residence address and CBVD mortality were included in the meta-analysis. The median quality score of the included studies was 90%. The results show the pooled HR of CBVD mortality per IQR increase in NDVI, indicating an overall reduction in the risk of CBVD mortality for each IQR increase in NDVI (HR (95% CI) = 0.96 (0.94, 0.97)) (Figure 4). NDVI buffers from the home address of each study were considered comparable by the authors, with all NDVI buffers ranging between 250 and 500 m from the residential address. The IQR of NDVI of the included studies ranged between 0.10 and 0.24. Sensitivity analyses were performed and showed no significant differences in the pooled results. The analysis of the funnel plot showed no evidence of publication bias. The results of the sensitivity analysis and funnel plots are reported in the Appendix A.

### 3.4. Other CVD Outcomes

Eight studies analyzing associations between UG and CVD morbidity were included. Six studies found a negative correlation between the amount of UG exposure and the risk of CVD morbidity, one did not find any statistically significant correlation, and one study found no statistically significant correlation between total UG exposure and CVD morbidity but found a negative correlation between green canopy exposure and CVD morbidity. Four studies analyzing associations between UG and IHD morbidity were included. Three studies found a negative correlation between the amount of UG exposure and the risk of IHD morbidity, and one found a positive correlation between tree canopy exposure and IHD morbidity. Three studies analyzing associations between UG and AMI morbidity were included and all three found a negative correlation between the amount of UG exposure and the risk of AMI morbidity (Table 1).

Moreover, one cohort study analyzed associations between tree canopy land cover and CVD or CVD hospital admission in type 2 diabetes patients, finding no statistically significant correlation for both outcomes. Associations between CVD hospital admission and tree canopy were also analyzed in an ecological study, which found no significant correlation between tree canopy land cover and CVD hospitalizations but found a negative correlation with tree density. Another cohort study found a negative correlation between NDVI and HF morbidity. Finally, one study analyzed correlations between UG exposure and CHD and stroke combined morbidity, finding no statistically significant results, but found an inverse correlation between UG exposure and CHD and stroke combined hospital admissions (Table 1).

### 3.5. Other CBVD Outcomes

Five studies analyzing associations between UG and stroke morbidity were included. Four studies found a negative correlation between the amount of UG exposure and the risk of stroke morbidity, and one found a positive correlation between tree canopy exposure and stroke morbidity (Table 1).

Furthermore, a cohort study evaluated associations between UG exposure and ischemic and hemorrhagic stroke, finding a negative correlation for both outcomes. Thus, an ecological study found no statistically significant correlation between UG exposure and ischemic or hemorrhagic stroke (Table 1).

### 3.6. Gender and Age Effects

Eight of the included studies performed stratified analysis by gender. Five of these studies found a protective effect of urban greenness on CVD outcomes that were statistically significant only in men. The other three studies found no statistically significant effect differences in males and females (Table 1).

Three studies included in the review performed stratified analysis by age. One study found an inverse correlation between UG exposure and CVD morbidity in individuals aged between 40 and 59, and for individuals aged more than 60, but loses the statistical significance for adults aged less than 40 years old. The other two studies found no effect modifications by different age groups (Table 1).

## 4. Discussion

### 4.1. Summary of the Results

The results of the present systematic review show that in most of the studies exposure to UG is correlated to a decreased risk of CVD and CBVD outcomes, suggesting an overall protective effect of UG on nearby residents’ cardiovascular health. The pooled results of the meta-analyses support this thesis, finding a reduction in CVD mortality, IHD mortality, and CBVD mortality in people living in areas with higher levels of NDVI.

This review highlights the lack of studies in low- and middle-income countries. Low- and middle-income countries are more susceptible to the dangerous effects of urbanization. The rate of urbanization and sprawl is higher in these territories than in others [49]. Furthermore, while CVD-related mortality has shown a dramatic decrease in countries with a high sociodemographic index (SDI) over the last decades, this cannot be applied to countries with lower SDI [50].

Several pathways explaining the association between UG exposure and reduced risk of CVD and CBVD outcomes are suggested in the existing literature. Residing in greener neighborhoods and living near parks and green areas were shown to incentivize physical activity [51]. Astell-Burt et al. found a positive association between residing in greener neighborhoods and a greater incidence of walking and moderate/vigorous physical activity [8]. Moreover, another study highlighted that the presence of parks in the residential neighborhood is correlated with an increase in cycling activity during leisure time [52].

Furthermore, UG spaces, and in particular the tree canopy, may represent a protective factor against heat stress. A lot of literature examples show how nature-based solutions (NBS) in urban architecture, from the street canopy to green roofs, have a beneficial effect in lowering the temperature in built environments [53,54]. NBS is defined by the EU Commission as “Solutions that are inspired and supported by nature, which are cost-effective, simultaneously provide environmental, social and economic benefits and help build resilience” [55]. This definition applied to the context of urban health refers to all those interventions of climate change adaptation that involve the use of green and/or blue infrastructures and may have a beneficial impact on health outcomes. A review showed that green roofs contribute to a median surface temperature reduction of 30 °C in cities with a hot–humid climate, and 28 °C in cities with a temperate climate [56]. A study simulated that a 50% increase in tree canopy cover over urban roads is correlated to a decrease in air temperature in urban street canyons of 4.1 K, a decrease in road-surface temperature of 15.4 K, and a reduction in building-wall temperature of 8.9 K [57].

Moreover, the impacts of UG exposure on reducing air pollution and traffic noise are reported in several studies [51,58,59,60]. Nature exposure is recognized to have a beneficial effect on mental health. [7,51] A systematic review showed inverse correlations between greenness exposure and both perceived and physiological stress indicators [61]. High temperature, air pollution, environmental noise, and chronic psychological stress represent well-known risk factors for CVD and CBVD [9,62]. Thus, the impact of UG on these factors may explain the association between greenness exposure and better cardiovascular and cerebrovascular health outcomes.

Different types of UG may play a role in understanding the explanatory mechanism of the results of this study. A cross-sectional study highlighted that total residential greenness was not associated with a statistically significant reduction in the risk of CVD, but residential tree canopy density in the same study population showed an inverse correlation [10]. Different types of UG areas in terms of usability may have different impacts on cardiovascular and cerebrovascular health. Public green spaces, such as parks or urban forests, are shown to incentivize physical activity and social aggregation [63,64]. The usability of those public spaces may vary in the presence or absence of sports or social facilities, perceived social safety, and maintenance state [65].

There is robust evidence in the existing literature supporting an association between the built environment and socio-economic disparities [66]. More deprived neighborhoods are most likely to be less green [67,68]. Thus, socio-economic variables, such as residents’ income, may represent an important confounding factor. The majority of the studies included in this review (n = 31) considered this aspect in their analysis, adjusting their models for socio-economic variables.

Since urban green environments are experienced differently by men and women [69], the association between UG exposure and CVD health may be subject to gender differences. Only eight studies included performed a stratified analysis by gender. Five of them found a statistically significant protective effect of urban greenness on CVD outcomes only in men [14,20,23,42,47], while the other three studies found no statistically significant effect differences in males and females [16,21,30]. These results suggest the existence of underlying gender-specific mediators on the cardiovascular health effects of UG. The different utilization of green facilities by men and women may be an explanation. A study found that public urban green area access is lower in women than men [70]. Moreover, self-perceived social safety may affect outdoor physical activities in women. Factors related to visibility, maintenance state, cleanliness, and the presence of viable facilities were shown to be more impactful on perceived safety in women than in men, suggesting UG design is an important mediator of gender differences in public green area use [71].

Only three of the included studies addressed age differences by performing a stratified analysis by age. One of them found an inverse correlation between UG exposure and CVD morbidity in the 40–59 years age group and for individuals aged over 60 [30], while the other two found no effect modifications according to age groups [42,48]. These results may be explained by the fact that the causes underlying CVD morbidity in younger adults are more likely to be genetic, non-modifiable factors [72].

### 4.2. Study Limitations

The main limitation of this review is the heterogeneity of the studies included. Different study designs, study populations, and choice of exposure variables restricted the quantitative comparability of the studies included. However, the heterogeneity in study settings may have excluded the possibility that the impact of UG exposure on cardiovascular and cerebrovascular health was a site-specific effect. Furthermore, the lack of general population-level experimental studies may limit the robustness of the results. Moreover, the lack of studies in low- and middle-income countries affects the generalizability of the results in such settings. Thus, further studies assessing the impacts of UG green exposure on CVD and CBVD in low- and middle-income settings should be prioritized and may have an important role in the policy-making process in these settings.

The studies included in the meta-analyses were matched by study design (cohort studies), choice of exposure variable (NDVI), and effect measure (HR per IQR of NDVI) but differed by study geographical settings. Thus, populations from different countries were compared in the meta-analyses. However, all studies took place in high-income countries (three in Canada, one in Italy, one in Belgium, and one in Switzerland), with a similar burden of CVDs and CBVDs [9]. Furthermore, statistical heterogeneity was high in the meta-analyses of CVD mortality and IHD mortality sub-groups (I2 = 97% and I2 = 90%, respectively), indicating that the pooled results should be interpreted with extreme caution. While random effect meta-analyses were performed in order to obtain more conservative pooled results, due to the high heterogeneity, the meta-analyses of CVD mortality and IHD mortality sub-groups should be interpreted as not conclusive, and the inferences from the results as being hindered. Sensitivity analyses of the IHD mortality sub-group meta-analysis showed a loss of statistical significance when the studies by Villeneuve et al. [18] or Vienneau et al. [27] were excluded.

Another limitation may be that NDVI was used as a UG exposure variable in most of the studies, including the ones in the meta-analyses. NDVI describes the difference between visible and near-infrared reflectance of vegetation, and it measures the density of green space coverage in an area [73]. Thus, NDVI does not describe the characteristics of green space, i.e., whether it is private or public, or whether it has a good or a bad maintenance state.

Finally, the impacts of UG exposure on other health outcomes were not analyzed in this systematic review. High levels of greenness were associated with possible beneficial effects on gestational, respiratory, and mental health outcomes [40,74,75]. On the other hand, vector-borne diseases (VBDs) may pose an issue, since some VBDs, such as tick-borne infections, are related to the presence of green environments [76].

## 5. Conclusions

Considering all the limitations described, the results of this systematic review suggest the existence of a beneficial effect of green exposure on cardiovascular and cerebrovascular health in urban settings. Further studies are needed to fully understand the explanatory mechanisms confirming the associations found in this review. General population-level experimental studies represent an opportunity to develop targeted future research projects aimed at analyzing the beneficial effects of UG-enhancing policies and the effects on human health. Furthermore, this study highlighted the necessity of re-thinking green areas in cities, reducing social disparities in the distribution of urban greenness, and designing more usable, safe, and gender-inclusive public urban green spaces. Policymakers should be aware of the fact that there is robust evidence of a correlation between UG exposure and lower risk of CVDs and CBVDs, incentivizing the creation of greener urban environments and considering the implementation and increase of UG areas as public health interventions.

## Figures and Tables

**Figure 1 ijerph-20-05966-f001:**
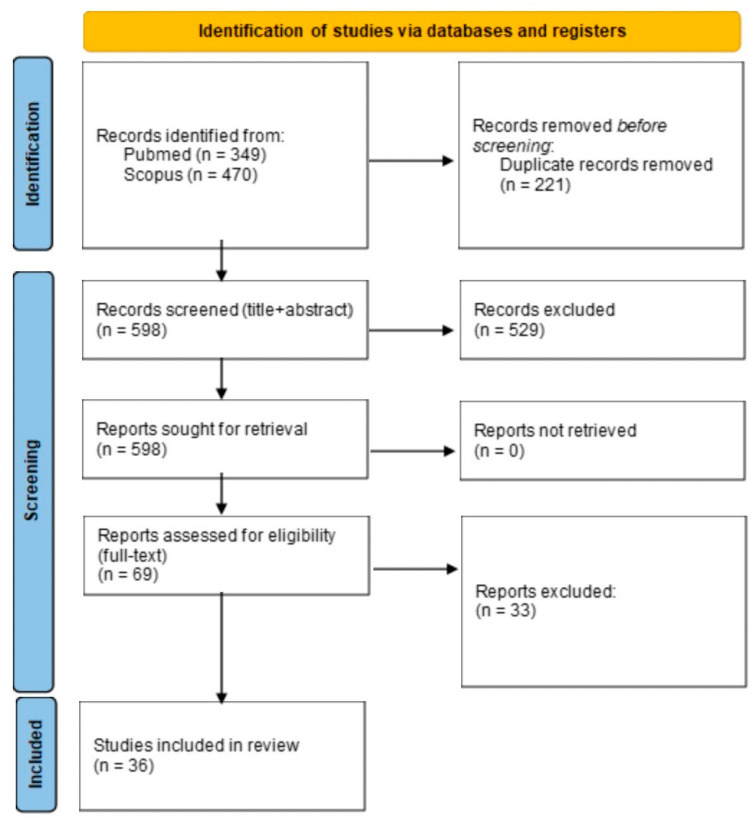
PRISMA flow diagram summarizing the process of article search and selection.

**Figure 2 ijerph-20-05966-f002:**
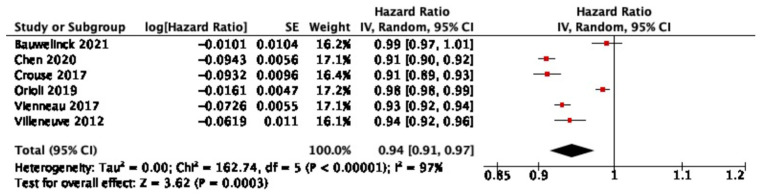
Results of the meta-analysis for CVD mortality. Studies included in the presented meta-analysis: Bauwelinck et al. [40], Chen et al. [37], Crouse et al. [25], Orioli et al. [29], Vienneau et al. [27], and Villeneuve et al. [18].

**Figure 3 ijerph-20-05966-f003:**
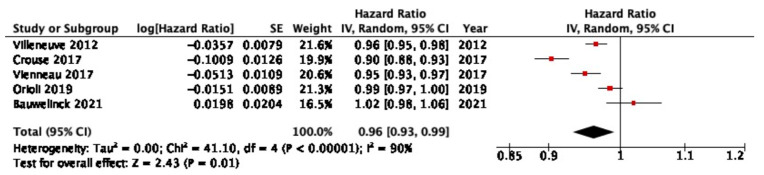
Results of the meta-analysis for IHD mortality. Studies included in the presented meta-analysis: Villeneuve et al. [18], Crouse et al. [25], Vienneau et al. [27], Orioli et al. [29], and Bauwelinck et al. [40].

**Figure 4 ijerph-20-05966-f004:**
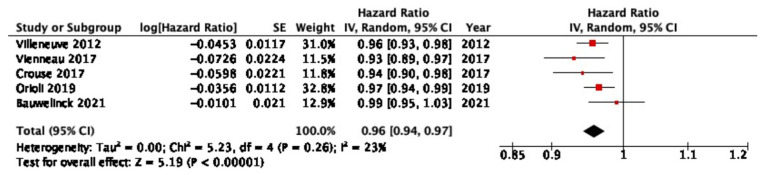
Results of the meta-analysis for CBVD mortality. Studies included in the presented meta-analysis: Villeneuve et al. [18], Vienneau et al. [27], Crouse et al. [25], Orioli et al. [29], and Bauwelinck et al. [40].

## Data Availability

All articles included in this systematic review are reported in Table 1.

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
