# Peer review of "Impacts of Urban Green on Cardiovascular and Cerebrovascular Diseases—A Systematic Review and Meta-Analysis"

_ijerph, 2023, doi:10.3390/ijerph20115966_

Round 1

Reviewer 1 Report (Previous Reviewer 1)

The manuscript has been significantly improved, there are no further comments.

Reviewer 2 Report (Previous Reviewer 2)

The authors had sufficiently revised the previous version and replied the comments. 

This manuscript is a resubmission of an earlier submission. The following is a list of the peer review reports and author responses from that submission.

Round 1

Reviewer 1 Report

Thank you for inviting me to evaluating the manuscript titled "Impacts of Urban green on Cardiovascular and Cerebrovascular diseases - A Systematic Review and Meta-analysis". The impact of Urban green on Cardiovascular and Cerebrovascular diseases was systematically summarized in detail in this article. I applaud the authors for their hard work, but the topic and content of the article are disappointing. The ambiguities and questions are the following but not limited to:

1. The quality assessment of the included literature is described too simply. Compared with the study of Gascon M et al., "Participants have been living at least 1 year in the studied area" was removed in this study, and the score range was still 0 to 100?

2. The number of studies included in each meta-analysis is relatively small.Whether the author consider increasing the number of databases.

3. In this study, heterogeneity was defined as I2 greater than 40 percent. Why? Please attach a reference basis.

4. In meta-analyses of urban green and CVD mortality or IHD mortality, I2 was not less than 90%. Do the results make sense in the context of such severe heterogeneity?

Author Response

We thank the Reviewer for his/her appreciation and very useful comments. All comments have been answered in detail point by point as it follows:

  1. “The quality assessment of the included literature is described too simply. Compared with the study of Gascon M et al., "Participants have been living at least 1 year in the studied area" was removed in this study, and the score range was still 0 to 100?”

We evaluated all the points proposed by Gascon et al. during the quality assessment of the studies included. The sentence was incorrect, so we rephrased it correctly. We thank the Reviewer for highlighting this error.

  1. “The number of studies included in each meta-analysis is relatively small. Whether the author consider increasing the number of databases.”

We thank the reviewer for the valuable comment, and we agree on the fact that the number of studies included in the meta-analyses were low. The number of studies included in the meta-analyses is relatively low due to the fact that we included only studies comparable for study design and method of assessment of the exposure, resulting in meta-analyses of cohort studies evaluating NDVI as exposure variable to urban green. Although we agree that expanding the number of databases may lead to a more extensive search, we evaluated that searching for epidemiological studies in two major, freely available for our institution, generalist databases (PubMed and Scopus) would be exhaustive in our specific case.

  1. “In this study, heterogeneity was defined as I2 greater than 40 percent. Why? Please attach a reference basis.”

We referred to the indications of the Cochrane Handbook for Systematic Reviews, being the reference to the sentence following the sentence discussed. The reference consecution was misleading, so we rephrase it in a more correct way. We kindly thank the Reviewer for pointing that out.

  1. “In meta-analyses of urban green and CVD mortality or IHD mortality, I2 was not less than 90%. Do the results make sense in the context of such severe heterogeneity?”

We strongly agree that in the presence of considerable heterogeneity, the results of a meta-analysis should be interpreted with caution. Although there is no consensus for an upper limit of heterogeneity, we agree that levels of I2over 90% represent a considerable high level of heterogeneity in the results of the studies included, determining a limitation to the meta-analytic results. Acknowledging that, we performed random-effect meta-analyses in order to obtain more conservative pooled results. However, we believe that in this case the two mentioned meta-analyses represent a useful quantitative summary of the results of the studies included, and, therefore, are worth being presented in the article. We also agree that this limitation should be discussed more thoroughly in the “study limitations” section, so we extended this part of the discussion (page 16, lines 229-233).

Reviewer 2 Report

This authors conducted a comprehensive survey and Meta-analysis on studies for the association between Urban green and cardiac-related diseases. The results basically coincide with most of the previous findings, alone with some extra information of gender as a potential modifier.  I have the following comments.

1. Abstract mentioned "Gender differences were found in four studies, with a protective effect of UG only statistically significant in men."

I consider this is an interesting result that make this paper more worthy for publication, indicating that the existence of gender-UG interaction effect on Cardiac-related disease. However this part only appears in discussion. I would like to see a result of meta-analysis on the interaction effect.   

2. On page 37 line 28-29, the authors stated:

Results show pooled HR of IHD mortality per IQR increase in NDVI, indicating no statistically significant reduction of risk of IHD mortality for each IQR increase in NDVI [HR (95% CI) = 0.96 (0.93, 0.99)] (Figure 3).

If both the bounds of CI <1, why it is " no statistically significant"?

3. The "weak (positive/negative) correlation" which the authors mentioned several times were not well defined. Barely showing CI is far from enough. Normally it is defined by the corresponsding p-value. 

4. In Table 1, "T3 vs T1" etc. in the column of "Comarison" was not defined, nor the "Q5 vs Q1" for Bixby et al. 2015

Author Response

We thank the Reviewer for his/her appreciation and very useful comments. All comments have been answered in detail point by point as it follows:

  1. “Abstract mentioned "Gender differences were found in four studies, with a protective effect of UG only statistically significant in men." I consider this is an interesting result that make this paper more worthy for publication, indicating that the existence of gender-UG interaction effect on Cardiac-related disease. However this part only appears in discussion. I would like to see a result of meta-analysis on the interaction effect.”

We thank the Reviewer for the suggestion and agree to consider the results of the gender differences analysis as an important and interesting part of the paper. So, we added an ‘ad hoc’ sub-section in the study results.

  1. “On page 37 line 28-29, the authors stated: Results show pooled HR of IHD mortality per IQR increase in NDVI, indicating no statistically significant reduction of risk of IHD mortality for each IQR increase in NDVI [HR (95% CI) = 0.96 (0.93, 0.99)] (Figure 3). If both the bounds of CI <1, why it is " no statistically significant"?”

The sentence was incorrect, so we rephrase it correctly. We thank the Reviewer for highlighting this error.

  1. “The "weak (positive/negative) correlation" which the authors mentioned several times were not well defined. Barely showing CI is far from enough. Normally it is defined by the corresponding p-value.”

We thank the Reviewer for this comment. The use of the word “weak” was erroneous in these contexts. Accordingly, we rephrased the sentences containing it and searched the entire document in order to prevent the presence of similar errors.

  1. “In Table 1, "T3 vs T1" etc. in the column of "Comparison" was not defined, nor the "Q5 vs Q1" for Bixby et al. 2015”

The abbreviations were not mentioned in the Table caption, resulting in an unclearness of the table content. So, we added an abbreviation section in the table caption.

Round 2

Reviewer 1 Report

The quality assessment of the included literature is described too simply, and the number of studies included in each meta-analysis is relatively small.

Author Response

We thank the Reviewer for his/her revision work. All comments have been answered in detail point by point as it follows:

  1. “The quality assessment of the included literature is described too simply, […]”

We thank the reviewer for the valuable comment. We described the quality assessment procedure more thoroughly in the Materials and Methods section (page 3). Furthermore, we added the reference to the article presenting the quality assessment score (Gascon et al.) at the end of the paragraph, to emphasize the fact that every detail about the score is well described in the original source. Moreover, we added the quality score of each included study, previously reported in the Supplementary material, to the main table (Table 1) in order to highlight the score reached by every article.

  1. “[…] and the number of studies included in each meta-analysis is relatively small.”

We thank the reviewer for the valuable comment, and we agree on the fact that the number of studies included in the meta-analyses were low. As described in the previous cover letter, the number of studies included in the meta-analyses is low due to our inclusion criteria for meta-analyses. Since only studies comparable for study design and method of assessment of the exposure were included, other studies found and included in the systematic review would not fit with our meta-analyses’ inclusion criteria. The resulting meta-analyses included only cohort studies evaluating NDVI as exposure variable to urban green. This reflects the high heterogeneity in study designs in the present literature on this topic.

Reviewer 2 Report

The authors replied the comment anf revised point-by point adequately, except that for the definition of "T3 vs T1".

The authors said that they had added the definition in Table cation, however it was nowhere to be seen.

Author Response

We thank the Reviewer for his/her appreciation and very useful comments. We also thank the reviewer for highlighting the absence of the Table 1 caption in the revised manuscript. We are sorry for the error. Accordingly, we added the table caption as intended in the first revision.